# Antibiotic and Nematocidal Metabolites from Two Lichen Species Collected on the Island of Lampedusa (Sicily)

**DOI:** 10.3390/ijms23158471

**Published:** 2022-07-30

**Authors:** Jesús García Zorrilla, Trifone D’Addabbo, Emanuela Roscetto, Chiara Varriale, Maria Rosaria Catania, Maria Chiara Zonno, Claudio Altomare, Giuseppe Surico, Pier Luigi Nimis, Antonio Evidente

**Affiliations:** 1Department of Chemical Sciences, University of Naples Federico II, Complesso Universitario Monte Sant’Angelo, Via Cintia 4, 80126 Napoli, Italy; evidente@unina.it; 2Allelopathy Group, Department of Organic Chemistry, Institute of Biomolecules (INBIO), School of Science, University of Cadiz, C/Republic Saharaui, s/n, 11510 Puerto Real, Spain; 3Institute for Sustainable Plant Protection-CNR, Unit of Bari, Via G. Amendola 122/d, 70126 Bari, Italy; trifone.daddabbo@ipsp.cnr.it; 4Department of Molecular Medicine and Medical Biotechnologies, University of Naples Federico II, Via Pansini 5, 80131 Napoli, Italy; emanuela.roscetto@unina.it (E.R.); chiaravarriale98@gmail.com (C.V.); mariarosaria.catania@unina.it (M.R.C.); 5Institute of Sciences of Food Production Sciences, National Research Council, Via Amendola 122/O, 70125 Bari, Italy; mariachiara.zonno@ispa.cnr.it (M.C.Z.); claudio.altomare@ispa.cnr.it (C.A.); 6Department of Agriculture, Food, Environment, and Forestry (DAGRI), Section of Agricultural Microbiology, Plant Pathology and Enthomology, University of Florence, Piazzale delle Cascine 18, 50144 Firenze, Italy; giuseppe.surico@unifi.it; 7Department of Life Sciences, University of Trieste, Via L. Giorgieri 10, 34127 Trieste, Italy; nimis@units.it

**Keywords:** lichen, *Ramalina implexa*, *Roccella phycopsis*, secondary metabolites, antibiotic and nematocidal activity

## Abstract

The antibiotic and nematocidal activities of extracts from two coastal lichen species collected on Lampedusa Island (Sicily), *Ramalina implexa* Nyl. and *Roccella phycopsis* Ach., were tested. Methyl orsellinate, orcinol, (+)-montagnetol, and for the first time 4-chlororcinol were isolated from *Roccella phycopsis*. (+)-Usnic acid was obtained from *Ramalina implexa*. The crude organic extract of both lichen species showed strong antibiotic activity against some bacterial species and nematocidal activity. Among all the pure metabolites tested against the infective juveniles (J2) of the root-knot nematode (RKN) *Meloydogine incognita*, (+)-usnic acid, orcinol, and (+)-montagnetol had significant nematocidal activity, comparable with that of the commercial nematocide Velum^®^ Prime, and thus they showed potential application in agriculture as a biopesticide. On the contrary, methyl orsellinate and 4-chlororcinol had no nematocidal effect. These results suggest that the substituent pattern at *ortho*-para-position in respect to both hydroxyl groups of resorcine moiety, which is present in all metabolites, seems very important for nematocidal activity. The organic extracts of both lichens were also tested against some Gram-positive and Gram-negative bacteria. Both extracts were active against Gram-positive species. The extract of *Ramalina implexa* showed, among Gram-negative species, activity against *Escherichia coli* and *Acinetobacter baumannii*, while that from *Roccella phycopsis* was effective towards all test strains, with the exception of *Pseudomonas aeruginosa.* The antimicrobial activity of (+)-usnic acid, methyl orsellinate, and (+)-montagnetol is already known, so tests were focused on orcinol and 4-chlororcinol. The former showed antibacterial activity against all Gram positive and Gram-negative test strains, with the exception of *A. baumannii* and *K. pneumoniae*, while the latter exhibited a potent antibacterial activity against Gram-positive test strains and among Gram-negative strains, was effective against *A. baumannii* and *K. pneumonia*. These results suggest, for orcinol and 4-chlororcinol, an interesting antibiotic potential against both Gram-positive and Gram-negative bacterial strains.

## 1. Introduction

Nature has always been an important source of new compounds with original carbon skeletons and biological activities applicable in different fields, but particularly in agriculture and medicine. Major sources have been terrestrial and marine higher organisms and microorganism, including plant pathogenic or endophytic microbes and marine microorganisms [1]. However, even if less investigated, lichens are also regarded as a potential source for novel biologically active natural compounds. Lichens represent the ecologically stable, self-supporting, and most successful symbiotic association between a photosynthetic partner (photobiont) such as an alga or a cyanobacterium, and a fungal partner (mycobiont) represented by a member of ascomycetes or basidiomycetes. Lichens include about 20,000 species worldwide, spread from arctic to tropical areas and from coastal areas to high mountains, and may occur in one of four main growth forms: crustose, squamulose, foliose and fruticose [2]. Presently, the terms ‘‘lichenized fungi’’ or ‘‘lichen-forming fungi’’ are often preferred to the term ‘‘lichens’’, and the classification of ‘‘lichens’’ is fully integrated with that of fungi. The large application of DNA sequencing to lichen taxonomy has resulted in rapid and significant changes, spanning from genus to family and order levels. Thus, two thirds of all the ascomycetes are lichenized, with a total number that is likely to exceed 28,000 [3].

Lichens produce >1000 specialized secondary metabolites, most of them not occurring in other organisms. Only a small number of such compounds have also been found in other fungi or higher plants. They mainly originate from the well-known acetate-malonate, shikimic acid, and mevalonic acid biosynthetic pathways [2]. Their isolation and identification have been carried out by allying techniques such as thin layer chromatography, high performance liquid chromatography, gas liquid chromatography, microcrystallization, physical and spectroscopic methods, and derivatization [4].

Lichen metabolites, such as zeorin, methylorsellinate methyl-□-orcinol carboxylate, methyl haematommate, lecanoric acid, salazinic acid, sekikaic acid, usnic acid, gyrophoric acid, and lobaric acid, have shown several and differentiated biological activities. They have shown promising antidiabetic and antioxidant properties, evidencing their potential for the treatment of diabetic mellitus and its associated complications, which in 2010 afflicted more than 285 million people world-wide with an increasing trend that is estimated to reach 439 million people affected by 2030 [5]. Moreover, lichen crude extracts or pure metabolites, such as usnic acid and its derivatives, lobaric acid, physodic acid, protolichesterinic acid, salazinic acid, atranorin, evernic acid, zeorin, diffractaic acid, psoromic acid, methyl β-orcinolcarboxylate methylorsellinate, and anziaic acid, have been shown to inhibit several enzymes, such as amylase, lipase, lipoxygenase, aromatase, cyclooxygenase, trypsin, β-glucoronidase, prolyl endopeptidase, monoamine oxidase, urease, tyrosinase, xanthine oxidase, thioredoxin reductase, glucosidase, topoisomerase, pancreatic elastase, phosphodiesterase, telomerase, and acetylcholinesterase. Thus, they were shown to possess potential as therapeutic agents for the inhibition of enzymes that are involved in some diseases or disorders, such as diabetic mellitus [5], Alzheimer’s disease, and obesity [2]. In addition, an extensive and in-depth investigation was also carried out on lichen metabolites belonging to other classes of natural compounds, which show promising anticancer activity [3].

The above literature reports prompt an extension of the investigation to newly discovered lichen species or to ecotypes of known species that grow in peculiar geographical areas or environments, with the aim to isolate and characterize new secondary metabolites with interesting chemical and biological properties.

Lampedusa, together with Linosa, form the Pelagie Islands Archipelago, minor Sicilian islands in the Mediterranean sea, which have proven to be a fertile source of microorganisms and plants producing secondary metabolites with noteworthy biological activities. Among these were the phytotoxins produced by *Neufusicoccum batangarum*, the causal agent of the scabby canker of cactus pear (*Opuntia ficus-indica* L.) [6], an exopolysaccharide produced as plant defense mechanism [7], and some farnesane-type sesquiterpenoids with antibiotic activity against resistant clinic bacteria that were isolated from the native plant *Chiliadenus lopadusanus* [8].

Thus, two strictly coastal lichens, the widespread *Roccella phycopsis* (right, Figure 1) and the rare *Ramalina implexa* (left, Figure 1), were collected from rocky and sloping coasts of Lampedusa Island.

This manuscript reports the isolation and chemical and biological characterization of the main metabolites isolated from both lichens *R. implexa* and *R. phycopsis,* focusing the biological investigation of these metabolites on the antibiotic and nematocidal activities, the latter here reported for the first time for any lichen metabolite.

## 2. Results and Discussion

The crude *n*-hexane and CH_2_Cl_2_ extracts from *Ramalina implexa* collected on Lampedusa Island were purified as reported in Section 3 section. From the *n*-hexane extract, only (+)-usnic acid was purified (**1**, Figure 2) while methyl orsellinate, orcinol, and (+)-montagnetol were obtained from the CH_2_Cl_2_ extract (**2**–**4**, Figure 2). The latter compounds are not typical of this species but have been found in *Roccella phycopsis*. We suppose that they are due to contamination from soredia of *Roccella*, the two species forming dense, entangled populations on steeply inclined, coastal rock faces. The compounds were identified comparing their physic and spectroscopic data with those reported in literature for **1** [9,10,11], **2** [12], **3** [13], and **4** [14]. These data were integrated from those obtained from ESIMS spectra. (+)-Usnic acid, methyl orsellinate, orcinol and (+)-montagnetol showed in their ESIMS spectra the protonated adduct ion at *m/z* 345, 183, 125, and 295, respectively. When the organic extracts obtained from *R. phycopsis* were purified with the same chromatographic procedures (see Materials and Methods), methyl orselllinate and orcinol were obtained from the *n*-hexane extract. Furthermore, the same two metabolites, (+)-montagnetol and 4-chlororcinol (**5**, Figure 2), were also isolated from CH_2_Cl_2_ organic extract. 4-Chlororcinol was identified by comparing its physical and spectroscopic properties with those reported in the literature [13]. Its ESIMS spectrum showed the protonated adduct ion at *m/z* 158.

Thus, the metabolic profile of the two lichen extracts differs for the production of 4-chloorcinol by *R. phycopsis* and the production of (+)-usnic acid by *R. implexa*. The latter species was reported to also contain salazinic acid [15]. Furthermore, *R. phycopsis* is also known to contain lecanoric acid as a major lichen compound [16]. These two acids were not isolated in the samples of both lichens collected in Lampeusa, forming the object of the present work, probably, as present in traces. Considering their strong acid polarity, another possibility could be that both acids should be extracted after a suitable acidification.

Usnic acid is the most common and abundant lichen metabolite [17]. It exhibited different biological activities, including antibiotic activity [17,18]. Antiproliferative activity against some cancer cell lines [19,20] and in vivo anti-inflammatory [21], antiviral [22], insecticidal [23,24], and phytotoxic activity [25,26] have also been described. Lately, given its pharmacological properties, the interest in the applicability in medicine for the development of antibacterial films [27] or as adjuvant to inhibit the Gram-negative bacteria *Acinetobacter baumannii* [28] has increased.

Methyl orsellinate is a metabolite close to orcinol, found in various lichen species [12,29,30,31] including *R. phycopsis* [32], fungi and mushrooms [33,34]. Its glucoside was also isolated from *Rhododendron primulaeflorum* [35]. Compound **2** showed antibiotic activity against *S. aureus* and *E. coli* [36] *Legionella* species [37], as well as antifungal activity [29,33], moderate cytotoxicity [38], and immunomodulatory [39] and enzyme inhibitory effects [18].

Orcinol is a phenolic compound found in various lichen species. It has also been reported in higher plants [13,40,41], fungi [42,43], and insects [44,45]. It has antiproliferative activity against cancer cells [20] and has shown acetylcholinesterase inhibition and antifungal properties against two *Cladosporium* species [46]. Although it has no nematocidal activity, some of its hemisynthetic derivatives have shown toxicity against *Meloidogyne javanica* [47].

(+)-Montagnetol has been previously found in different *Roccella* species, including *R. phycopsis* [48] and *R. montagnei* [21]. The study by Basset et al. (2009) [49] confirmed its stereochemistry synthetizing both epimers. Thus, the natural montagnetol was assigned as the (+)-epimer. Compound **4** showed antibacterial and antifungal activities against *P. aeruginosa*, *S. aureus*, *E. coli* and *C. albicans* [14], and antioxidant activity [50].

4-Chlororcinol is a chlorinated derivative of orcinol found in higher plants [13] and fungi [51]. It was found only in traces in *Lecanora straminea* [52]. Thus, this is the first report on 4-chlororcinol as metabolite of *R. phycopsis* [32]. It showed significant phytotoxic and zootoxicity activity [51] as well as anti-inflammatory activity in vivo [53], and growth inhibition of human tubercle bacilli in vitro [54].

The metabolites **1**–**5** were assayed for antibiotic and nematocidal activity. When assayed against the infective juveniles (J2) of the root-knot nematode (RKN) *Meloydogine incognita* at concentration 1 mg/mL, the compounds **1**, **3** and **4** had significant nematocidal activity, comparable with that of the commercial nematocide Velum^®^ Prime (Figure 3). On the contrary, compounds **2** and **5** had no nematocidal effect. The percent mortality of nematode J2 treated with compounds **1**, **3**, and **4** were 88.7%, 89.9%, and 97.1%, respectively, while the treatment with the commercial nematocide Velum^®^ Prime at the practical application dose resulted in 94.6% mortality of J2s. The result obtained testing orcinol is in agreement with that previously reported testing **3** on the other species, *M. javanica* [47]. RKNs of genus *Meloidogyne* are annually responsible of 12–15% of world crop losses [55], though often underestimated because of their microscopical size and the non-specific symptoms of their attacks. The increasingly severe restrictions on the use of synthetic pesticides are impeding the search for new sources of safe nematocidal products, including microbial metabolites such as those investigated in this study, whose nematocidal activity is herein reported for the first time.

The results reported in Figure 3 show that the nematocidal activity depends on the substitution pattern of the resorcine moiety present in all five metabolites. The substituent at *ortho-para*-position in respect to both hydroxyl groups of resorcine seems very important for the activity while the different groups present on the other position seem to have less importance.

The antibacterial potential of extracts from *R. phycopsis* and *R. implexa* was evaluated against methicillin-resistant *Staphylococcus aureus* (MRSA) ATCC 43300, *Enterococcus faecalis* ATCC 29212, *Escherichia coli* ATCC 25922, *Acinetobacter baumannii* BAA747, *Pseudomonas aeruginosa* ATCC 27853, and KPC carbapenemase producer *Klebsiella pneumoniae* BAA1705. Both extracts were effective against Gram-positive test strains. Among Gram-negative bacteria, the extract from *R. implexa* showed activity against *E. coli* and *A. baumannii*, while that from *R. phycopsis* was effective towards all test strains, with the exception of *Pseudomonas aeruginosa*. As shown in Table 1, the total inhibition of bacterial growth was obtained at the highest tested concentration of 1000 μg/mL for *R. phycopsis* and at the concentration of 500 μg/mL for *R. implexa*.

Among the metabolites obtained from the purified extracts, (+)-usnic acid, methyl orsellinate, and (+)-montagnetol are already known for their antimicrobial activity, while orcinol and 4-chlororcinol were selected to test their antibacterial potential given the paucity of data in the literature. The results are reported in Table 2. 

Orcinol exhibited antibacterial activity against all Gram-positive and Gram-negative test strains, with the exception of *A. baumannii* and KPC producer *K. pneumoniae*. Minimum inhibitory concentration (MIC) values ranged from 300 to 18.75 μg/mL, with the lowest value recorded for MRSA. Minimum bactericidal concentration (MBC) values exceeded the concentration range we tested with the exception of *E. coli* for which an MBC of 37.5 μg/mL was found. 4-Chlororcinol exhibited a potent antibacterial activity against Gram-positive test strains with MIC values of 1.17 μg/mL for MRSA and 75 μg/mL for *E. faecalis*. Regarding Gram-negative strains, 4-chlororcinol was effective against *A. baumannii* and KPC *K. pneumoniae* with MIC values of 300 μg/mL. The lowest MBC value, corresponding to 4 μg/mL, was found for *S. aureus*, while the MBC values were greater than 300 μg/mL for all other strains.

To date, only the activity of orcinol against *Cladosporium-**species* [46] and 4-chloro rcinol on *Mycobacterium tuberculosis* [54] has been reported. In our study, these compounds were tested against pathogens belonging to the ESKAPE group, which includes both Gram-negative and Gram-positive bacteria that represent a leading cause of nosocomial infections [56]. Such pathogens frequently exhibit multidrug resistance profiles that limit treatment options. Indeed, the emergence and continuous spread of multidrug resistant bacteria that threaten public health throughout the world drive the need for novel therapeutic options. The activity of orcinol and 4-chlororcinol against Gram-positive and Gram-negative strains makes these compounds very interesting as potential antimicrobials.

## 3. Materials and Methods

### 3.1. General Experimental Procedure

The optical rotations were recorded on a P-1010 digital Jasco polarimeter (Tokyo, Japan) in CH_2_Cl_2_ unless otherwise noted. IR spectra were recorded as glassy film on a Perkin Elmer Spectrum 100 FT-IR spectrometer (Milan Italy) while the UV and ECD spectra were recorded at room temperature on a JASCO (Tokyo, Japan) J1500 spectropolarimeter, using 0.5-mm cells and concentrations of 4.7 mM in acetonitrile. ^1^H NMR spectra were recorded in CDCl_3_, CD_3_OD, or CD_3_COCD_3_, also used as internal standards, at 400 MHz on a Bruker (Karlshrue, Germany) spectrometer. HRESI and ESI mass spectra and liquid chromatography (LC)/MS analyses were carried out using the LC/MS TOF system Agilent 6230B (Agilent Technologies, Milan, Italy), HPLC 1260 Infinity. A Phenomenex (Bologna, Italy) LUNA (C18 (2) 5u 150 × 4.6 mm column) was utilized to perform the HPLC separations, using an isocratic elution with CH_3_CN-H_2_O (85:15) at flow rate of 1 ml/min; 10 μL of sample was injected using a 20 μL loop. The spectra were recorded in positive mode in low resolution. Preparative and analytical TLC were performed on silica gel (Merck, Kieselgel 60 F_254_, 0.50 and 0.25 mm, respectively) plates (Merck, Darmstadt, Germany), while column chromatography was performed on silica gel (Merck, Kieselgel 60, 0.063–0.200 364 mm). The spots were visualized by exposure to UV light and/or iodine vapors. Sigma-Aldrich Co., (Milan, Italy) supplied all the reagents and the solvents. 

### 3.2. Lichens

Whole samples of *Ramalina implexa* and *Roccella phycopsis* were collected in August 2021 on Lampedusa Island (Italy) by Mr. Fabio Giovanetti and Prof. G. Surico, University of Florence, Italy. Samples of the two lichens were examined for species identification by prof. Pier Luigi Nimis. Specimens of both lichens were deposited in the collection of the Department of Agriculture, Food, Environment, and Forestry (DAGRI), Section of Agricultural Microbiology, Plant Pathology and Enthomology, University of Florence, Italy, DAGRI, and in the TSB lichen herbarium.

### 3.3. Extraction and Purification of Metabolites from the R. implexa Lichen Species

The dried and powdered thallus of lichen *R. implexa* (250 g) was blended in 2 L of MeOH/Milli-Q water (1% NaCl) and macerated under stirring overnight. The sample was centrifuged at 7000 rpm at 5 °C for 30 min, and the resulting aqueous phase (1 L) extracted in *n*-hexane (3 × 500 mL) and CH_2_Cl_2_ (3 × 500 mL). The residue solid was extracted again following the same procedure. The organic extracts were combined, dried (Na_2_SO_4_), filtered and evaporated under reduced pressure. A residue of 87.6 and 391.1 mg were obtained from *n*-hexane and CH_2_Cl_2_ extracts, respectively. The *n*-hexane extract was fractioned by column chromatography, eluted with CHCl_3_, obtaining four groups of homogeneous fractions F1–F4. The residue (69.0 mg) of F4 was further purified by preparative TLC using CHCl_3_-*i*-PrOH (98:2) yielding 14.0 mg of a homogeneous compound identified as below reported as (+)-usnic acid (**1**). The CH_2_Cl_2_ extract was fractioned by column chromatography, eluted with CHCl_3_-*i*-PrOH (gradient from 9:1 to 7:3), obtaining five groups of homogeneous fractions FF1-FF5. The residue of FF2 (23.5 mg) was further purified by preparative TLC eluted with *n*-hexane-EtOAc (6:4), affording a homogeneous compound (11.2 mg) identified as below reported as methyl orsellinate (**2**). The residue FF3 (58.9 mg) was further purified by preparative TLC eluted with CHCl_3_-*i*-PrOH (9:1) obtaining a homogeneous compound (36.1 mg) identified as below reported as orcinol (**3**). The residue FFF5 (13.6 mg) was further purified by analytical TLC eluted with CHCl_3_-*i*-PrOH (8:2) yielding a homogeneous compound (4.4 mg) identified as below reported as (+)-montagnetol (**4**).

#### 3.3.1. (+)-Usnic acid (**1**)

Compound **1** had: [α]^25^_D_ +372° (*c* 1.1, CH_2_Cl_2_) ([α]^25^_D_ +422° (*c* 0.1, CH_2_Cl_2_) lit. 9). ^1^HNMR spectrum (see Appendix A) very similar to that previously reported [10,11]; ESIMS (+) *m/z*: 345 [M + H]^+^ (see Appendix A).

#### 3.3.2. Methyl orsellinate (**2**)

Compound **2** had: ^1^H NMR spectrum (see Appendix A) very similar to that previously reported [18]; ESIMS (+) *m/z*: 183 [M +H]^+^ (see Appendix A).

#### 3.3.3. Orcinol (**3**)

Compound **3** had: ^1^H NMR spectrum (see Appendix A) very similar to those previously reported [13]; ESIMS (+) *m/z*: 125 [M + H]+ (see Appendix A).

#### 3.3.4. (+)-Montagnetol (**4**)

Compound **4** had: [α]^25^_D_ +14° (*c* 0.5, Me_2_CO) ([α]^20^_D_ +16.8° (*c* 0.4, EtOH), lit. 14. ^1^H NMR spectrum (see Appendix A) very similar to those previously reported (lit. 14); ESIMS (+) *m/z*: 295 [M + Na]^+^ (see Appendix A).

### 3.4. Extraction and Purification of Metabolites from R. phycopsis

Dried and powdered thalli of *R. phycopsis* (107 g) were blended in 0.8 L of MeOH/Milli-Q water (1% NaCl) and macerated under agitation overnight. The sample was centrifuged at 7000 rpm at 5 °C for 30 min, and the resulting aqueous phase (0.4 L) extracted in n-hexane (3 × 200 mL) and then with CH_2_Cl_2_ (3 × 200 mL). The solid residue was extracted again following the same above-described procedure. Both organic extracts were dried (Na_2_SO_4_), filtered and evaporated under reduced pressure giving two oily residues of 59.0 and 744.1 mg for *n*-hexane and CH_2_Cl_2_ extracts, respectively. The *n*-hexane extract was purified by preparative TLC using CHCl_3_-*i*-PrOH (98:2), yielding methyl orsellinate (**2**, 3.9 mg) and orcinol (**3**, 0.6 mg). The CH_2_Cl_2_ extract was fractioned by column chromatography in CHCl_3_-*i*-PrOH (gradient from 9:1 to 7:3), obtaining ten groups of homogeneous fractions, F1–F10. The residues of F2 (2.59 mg) and F3 (69.7 mg) were combined and further purified by two successive steps on TLC eluted with *n*-hexane-EtOAc (6:4) yielding methyl orsellinate (**2**, 53.9 mg) as homogeneous compound. The residue of FF4 (7.3 mg) was further purified by TLC eluted with *n*-hexane-EtOAc (7:3), yielding a homogeneous compound (1.60 mg) identified as below reported as 4-chlororcinol (**5**). The residues of F5 (302, 6 mg) and F6 (122, 5 mg) were further purified by column chromatography in CHCl_3_-*i*-PrOH (9:1), affording orcinol (**3**, 118 mg) as homogeneous compounds. The residue of FF10 (32.2 mg) was further purified by preparative TLC eluted with CHCl_3_-*i*-PrOH (8:2), obtaining (+)-montagnetol (**4**, 18.8 mg) as pure compound.

#### 4-Chlororcinol (**5**)

Compound **5** had: ^1^H NMR spectrum (see Appendix A) very similar to those previously reported [13]; ESIMS (+) *m/z*: 159 [M + H]^+^ (see Appendix A).

### 3.5. Nematocidal Activity

The nematocidal effects of the pure compounds **1**–**5** were tested on second-stage juveniles (J2) of the RKN *M. incognita*. Formed egg masses of an Italian population of *M. incognita* were handpicked from infested roots of tomato plants (cv. Regina di Fasano) grown in a glasshouse at a 25 ± 2 °C constant temperature. The egg masses were then incubated in distilled water in a growth chamber at 25 °C and the emerged infective juveniles were collected and stored in water at 5 °C until their use. A 0.5 mL volume of a water suspension of juveniles, containing about 150 specimens, was poured into 1.5 mL Eppendorf vial, which were then supplemented with 10µL of a 50 µg/mL solution of the test compounds in methanol or 10 µL of *R. implexa* lichen crude extract as reference. Distilled water, 2% methanol, and a 0.625 mL/L water solution of a liquid formulation of the nematocide Velum^®^ Prime (41.5% *w/w* of the active principle fluopyram, Bayer Crop Science Italia, Milan, Italy) were included as controls. Four replicates were prepared for each treatment and control. Juveniles were incubated in a growth chamber at 25 °C, over a 96-h time lapse, after which a drop of 1N NaOH was added to the nematode suspension to cause motion of viable J2, while the dead J2s stayed still. Dead specimens were counted under a dissecting microscope and the percent mortality was calculated according to Abbott’s formula [57]:m = 100 × (1 − nt/nc)(1)
where m = percent immobility/mortality; nt = number of J2 still mobile/viable after the treatment; nc = number of mobile/viable J2 in the water control.

### 3.6. Statistical Analysis

Data of J2 mortality were statistically analyzed by one-way ANOVA and the means were compared with the Fisher’s least significant difference test (*p* ≤ 0.05) using the software PlotIT 3.2 (Scientific Programming Enterprises, Haslett, MI, USA).

### 3.7. Antibiotic Activity

The antibacterial activity of lichen extracts and secondary metabolites was assayed on two Gram-positive reference strains: methicillin-resistant *Staphylococcus aureus* (MRSA) ATCC 43,300 and *Enterococcus faecalis* ATCC 29212, and four Gram-negative reference strains: *Escherichia coli* ATCC 25922, *Acinetobacter baumannii* BAA747, *Pseudomonas aeruginosa* ATCC 27853, and *Klebsiella pneumoniae* BAA1705. All strains were stored as 15% (*v/v*) glycerol stocks at −80 °C. Before each experiment, cells were subcultured from the stocks onto TSA plates at 37 °C for 24 h. The strains were identified by MS MALDI-TOF (Bruker Daltonics, Bremen, Germany) and their antibiotic susceptibility profiles were evaluated by Vitek 2 (bioMérieux, Marcy-l’Étoile, France).

The antibacterial potential of lichen extracts and two purified compounds, orcinol and 4-chlororcinol, was assessed through standard microdilution assay in 96 well polystyrene plates using brain heart infusion (BHI) broth. Bacterial suspensions with a turbidity of 0.5 McFarland (corresponding to 1–5 × 10^8^ cells/mL) were prepared for each test strain. The inoculum was diluted in BHI-broth to obtain a suspension of 5 × 10^6^ CFU/mL. A volume of 100 μL of the inoculum was added to each well, obtaining a final density of 5 × 10^5^ CFU/well. The antibiotic activity was tested by adding 100 μL of serial dilutions (1:2) of test substances to the wells starting from 1000 μg/mL for lichen extracts and 300 μg/mL for orcinol and 4-chlororcinol. The wells without substances were used as a positive growth control. Teicoplanin (ranged from 0.06 to 4 μg/mL) and colistin (ranged from 0.5 to 32 μg/mL) were used as control conventional antibiotics for Gram-positive and Gram-negative, respectively. Serial dilutions of compound solvent (starting from 1% dimethyl sulfoxide) were tested to be sure that it did not act on the bacterial growth. Thereafter, the plates were incubated at 37 °C under aerobic conditions for 20 h. The minimal inhibitory concentration (MIC) and minimal bactericidal concentration (MBC) of test substances were determined: the MIC was defined as the lowest concentration of substance that caused no visible bacterial growth in the wells. The MBC was defined as the lowest concentration of substance that kills the cells in the planktonic culture. The extracts and each compound were tested in triplicate and each experiment was performed twice.

## 4. Conclusions

The crude organic extracts of both lichen species showed strong antibiotic activity against some bacterial species and nematocidal activity. (+)-Usnic acid, orcinol, and (+)-montagnetol had significant nematocidal activity against *Meloydogine incognita* comparable with that of the commercial nematocide Velum^®^, and thus potential application in agriculture as biopesticides. As methyl orsellinate and 4-chlororcinol had no nematocidal effect, it is possible to hypothesize a role in affecting this activity of the substituent pattern at the *ortho-para*-position in respect to both hydroxyl groups of the resorcine moiety present in all the metabolites. The organic extracts of both lichens were also tested against some Gram-positive and Gram-negative bacteria. As the antimicrobial activity of (+)-usnic acid, methyl orsellinate, and (+)-montagnetol is already known, the tests were focused on orcinol and 4-chlororcinol. The first one showed antibacterial activity against all Gram-positive and Gram-negative test strains, with the exception of *A. baumannii* and *K. pneumoniae*, while 4-chlororcinol exhibited potent antibacterial activity against Gram-positive test strains and among Gram-negative strains, was effective against *A. baumannii* and *K. pneumoniae*. These results suggest that orcinol and 4-chlororcinol as an interesting potential antibiotic against both Gram-positive and Gram-negative bacterial strains. The metabolites showing potential as nematocide and/or antibiotic activity against human pathogen should be analyzed for their toxicological activity before evaluating their potential practical application in agriculture and/or medicine. Moreover, a most convenient method for their production at large scale as well as the moist suitable formulation to increase their efficacy should be developed.

## Figures and Tables

**Figure 1 ijms-23-08471-f001:**
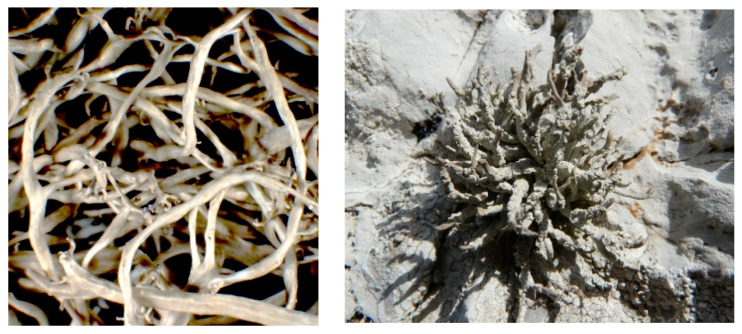
*Ramalina implexa* (left, photo P.L. Nimis, TSB 5975) and *Roccella phycopsis* (right, photo A. Moro).

**Figure 2 ijms-23-08471-f002:**
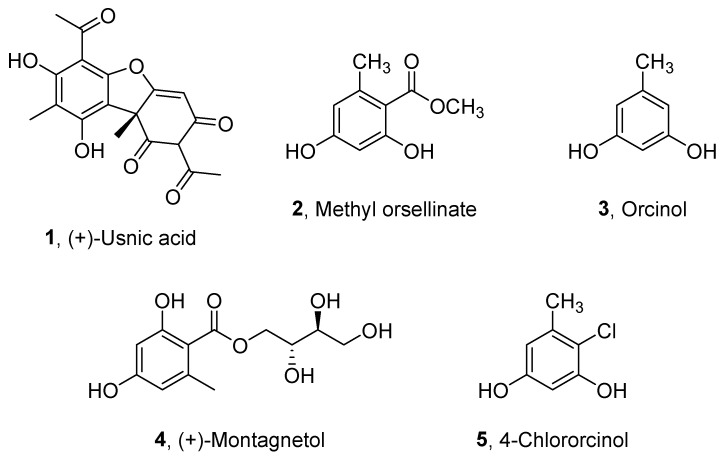
Metabolites isolated from *Ramalina implexa* and *Roccella phycopsis*: (+)-usnic acid, methyl orsellinate, orcinol, (+)-montagnetol and 4-chlororcinol (**1**–**5**).

**Figure 3 ijms-23-08471-f003:**
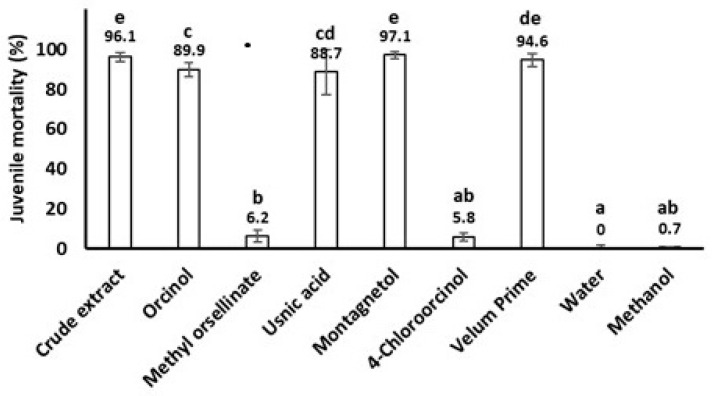
Nematocidal activity of the lichen metabolites usnic acid (**1**), methyl orselinate (**2**), orcinol (**3**), montagnetol (**4**) and 4-chlororcinol (**5**) against the root-knot nematode *Meloydogine incognita*, compared to the commercial nematocide Velum^®^ Prime. Data shown are percent mortalities of second stage juveniles (J2) ± standard deviations. Bars with different letters are significantly different (*p* = 0.05) according to one-way ANOVA and LSD test.

**Table 1 ijms-23-08471-t001:** MIC values of lichen extracts against test reference strains.

Bacterial Species	Strain Number	*Roccella phycopsis* Extract MIC (µg/mL)	*Ramalina implexa* Extract MIC (µg/mL)	Control Antibiotic MIC (µg/mL)
MRSA	ATCC 43300	1000	500	1
*Enterococcus faecalis*	ATCC 29212	1000	500	≤0.5
*Escherichia coli*	ATCC 25922	1000	500	1
*Acinetobacter baumannii*	BAA 747	1000	500	0.78
*Pseudomonas aeruginosa*	ATCC 27853	n.d	n.d	4
*Klebsiella pneumoniae* KPC	BAA 1705	1000	n.d	<2

Conventional antibiotics teicoplanin (TE) and colistin (CO) were used as positive control of activity on Gram-positive and Gram-negative, respectively. n.d. = not detected; MRSA = methicillin-resistant *S. aureus*; KPC = *K. pneumoniae* carbapenemase.

**Table 2 ijms-23-08471-t002:** MIC and MBC values of orcinol and 4-chlororcinol against test reference strains.

ATCC Bacterial Strains	Orcinol MIC/MBC (μg/mL)	4-Chlororcinol MIC/MBC (μg/mL)	Control Antibiotic MIC (µg/mL)
MRSA 43300	18.75/ > 300	1-17/4	1
*E. faecalis* 29212	9.37/ > 300	75/ > 300	≤0.5
*E. coli* 25922	9.37/37.5	n.d	1
*A. baumannii* 747	n.d	300/ > 300	0.78
*P. aeruginosa* 27853	300/ > 300	n.d	4
*K. pneumoniae* 1705	n.d	300/ > 300	<2

Conventional antibiotics teicoplanin (TE) and colistin (CO) were used as positive control of activity on Gram-positive and Gram-negative, respectively. n.d. = not detected.

## Data Availability

Data are contained within the text.

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
