# Peer review of "Antibiotic and Nematocidal Metabolites from Two Lichen Species Collected on the Island of Lampedusa (Sicily)"

_ijms, 2022, doi:10.3390/ijms23158471_

Round 1

Reviewer 1 Report

The paper titled “Antibiotic and nematocidal metabolites from two lichen species collected on the island of Lampedusa (Sicily)” deals with important topic of antimicrobial acitivity of orcinol and 4-chlororcinol against both gram-positive and gram-negative bacterial strains. The paper could be interested for IJMS readers and it is well written, but it needs some improvements and modifications, listed below:

1)      Introduction section: please describe the aim of the work more clearly

2)    The use of the number for each specific metabolite isolated throughout the text is redundant.

3)      Figure 2 and line 226. Please standardize the spelling of 4-chlororcinol in the figure, in the figure legend and throughout the text.  

4)      Line 281: 3 x 500ml, the unit of measure is missing throughout the Materials and Methods section

5)      In 3.1 paragraph “General Experimental Procedure” please describe the condition of LC/MS TOF and the eluents used. The analysis was done in isocratic o gradient?

6)      Line 355: How was done the statistical analysis? What kind of software was used? Please separate a paragraph with the statistical analysis

7)      Line 378-379: “ranged from..” requires “from … to..”, otherwise rewrite the sentence

8)      Line 413: review the spelling “Not applicanle” change with “Not applicable”

9)      Conclusions section: please describe the possible further studies due to this finding

10)   A general revision of English language is suggested

Reviewer 2 Report

Dear Authors,

The title of the article sounds interesting. Also, performed analyses  look interesting.

The article does not require significant corrections. 

The introduction is comprehensive and introduces well the research topic,

Methodology described in detail – reproducible

The experimentum design and data analysis are appropriate.

In my opinion this work should make a good contribution to the literature. Further more, in my opinion, the paper will attract a wide readership. The MS looks good. Therefore, I have no serious substantive comments to the MS.

Recommendations for authors:

However, the authors need to correct the technical shortcomings:

1.The chart does not match the whole manuscript, please change the font, reduce it a bit. Chart should be flush with paragraphs

2. You should format tables and captions so that they are flush with paragraphs

3. Please read the MS once more and correct any minor shortcomings, e.g. punctuation, etc.

Round 2

Reviewer 1 Report

I would like to thank the Authors for taking into account the Reviewer's suggestions.

However, in the 3.1 paragraph some details are missing. The authors declare that the analysis were carried out using the LC/MS TOF. Please specify more clearly the chromatographic conditions (Flow rate, the injection volume, etc..) and the mass spectrometric conditions (the mass resolution, the MS experiments were conducted both positive and negative ion modes?  The analytical conditions etc..)
